# A Markerless Gene Deletion System in *Streptococcus suis* by Using the Copper-Inducible *Vibrio parahaemolyticus* YoeB Toxin as a Counterselectable Marker

**DOI:** 10.3390/microorganisms9051095

**Published:** 2021-05-19

**Authors:** Chengkun Zheng, Man Wei, Jun Qiu, Jinquan Li

**Affiliations:** 1Joint International Research Laboratory of Agriculture and Agri-Product Safety, The Ministry of Education of China, Yangzhou University, Yangzhou 225009, China; fhzxmwei@163.com (M.W.); qiujun527@163.com (J.Q.); 2Jiangsu Key Laboratory of Zoonosis, Yangzhou University, Yangzhou 225009, China; 3College of Food Science and Technology, Huazhong Agricultural University, Wuhan 430070, China; lijinquan@mail.hzau.edu.cn

**Keywords:** *Streptococcus suis*, markerless gene deletion, toxin, YoeB, counterselectable marker

## Abstract

*Streptococcus suis* is an important zoonotic pathogen causing severe infections in swine and humans. Induction of the *Vibrio parahaemolyticus* YoeB toxin in *Escherichia coli* resulted in cell death, leading to the speculation that YoeB*_Vp_* can be a counterselectable marker. Herein, the counterselection potential of YoeB*_Vp_* was assessed in *S. suis*. The *yoeB_Vp_* gene was placed under the copper-induced promoter P*copA*. The P*copA*-*yoeB_Vp_* construct was cloned into the *S. suis-E. coli* shuttle vector pSET2 and introduced into *S. suis* to assess the effect of YoeB*_Vp_* expression on *S. suis* growth. Reverse transcription quantitative PCR showed that copper induced *yoeB_Vp_* expression. Growth curve analyses and spot dilution assays showed that YoeB*_Vp_* expression inhibited *S. suis* growth both in liquid media and on agar plates, revealing that YoeB*_Vp_* has the potential to be a counterselectable marker for *S. suis*. A SCIY cassette comprising the spectinomycin-resistance gene and copper-induced *yoeB_Vp_* was constructed. Using the SCIY cassette and peptide-induced competence, a novel two-step markerless gene deletion method was established for *S. suis*. Moreover, using the Δ*perR* mutant generated by this method, we demonstrated that PmtA, a ferrous iron and cobalt efflux pump in *S. suis*, was negatively regulated by the PerR regulator.

## 1. Introduction

*Streptococcus suis* is a Gram-positive, facultative anaerobe that threatens the swine industry and public health worldwide [1]. It is responsible for various swine diseases, including meningitis, septicemia, pneumonia, endocarditis, and arthritis [2]. Generally, *S. suis* is considered one of the most important bacterial pathogens that lead to significant economic losses to the swine industry [3]. Indeed, a recent survey revealed that its isolation rate was 16.9%, ranking first among the bacterial pathogens isolated from Chinese pig farms from 2013 to 2017 [4]. More seriously, *S. suis* can be transmitted to humans by minor skin injuries or the gastrointestinal tract, causing meningitis, streptococcal toxic shock-like syndrome, and some other clinical symptoms [5]. In 1968, the first human case of *S. suis* infection was described in Denmark; since then, over 1600 human cases have been reported worldwide by the end of 2013, some of which were fatal [6]. Remarkably, two great outbreaks of human *S. suis* infections occurred in China in 1998 and 2005, resulting in 25 cases with 14 deaths and 215 cases with 39 deaths, respectively [7,8]. In recent years, *S. suis* still frequently caused sporadic human cases worldwide [9,10,11,12,13,14].

Over the past few decades, significant progress has been made toward understanding the physiology and pathogenesis of *S. suis*. A number of virulence-related factors have been described in *S. suis* [15,16]. Recently, *in vivo* transcriptomes and transposon mutant libraries have been applied to identify genes involved in the virulence traits of *S. suis* [17,18,19]. Usually, studies related to the physiology and pathogenesis of *S. suis* rely on gene deletion mutants. In *S. suis*, the most frequently used gene deletion system is the pSET4s thermosensitive suicide vector [20]. For gene deletion using pSET4s, a knockout vector is constructed and introduced into the wild-type (WT) *S. suis* strain by electroporation; subsequently, the mutant is selected after two steps of allelic exchange. As this system contains no counterselectable marker, the mutant must be picked out from many potential colonies. In addition, electrotransformation does not work well for certain *S. suis* isolates [21]. Except for allelic exchange using the pSET4s vector, a cloning-independent method employing peptide-induced competence has been established in *S. suis* [22]. This method allows high-throughput mutation; however, the mutant carries a spectinomycin resistance gene, limiting its vaccine potential. Only recently, Zhu et al. developed a markerless gene deletion method in *S. suis* Chz serotype with the utilization of the ComRS system and sucrose sensitivity [21].

Toxin-antitoxin (TA) systems are small genetic modules widely distributed in the plasmids or chromosomes of bacteria and archaea [23]. Typically, they are composed of a gene encoding a stable toxin and a gene encoding an unstable antitoxin [24,25]. Under stress conditions, toxins are released from the TA complex and target various cellular functions to inhibit cell growth, making them valuable for counterselection [26,27,28,29,30]. In a previous study, we identified a chromosomal type II toxin-antitoxin system, YefM-YoeB, in *Vibrio parahaemolyticus*; induction of YoeB*_Vp_* in *Escherichia coli* resulted in cell death [31]. This result has led us to speculate that YoeB*_Vp_* can be a counterselectable marker for *S. suis*.

In this study, the YoeB*_Vp_* toxin was tested for the counterselection potential in *S. suis*. Using YoeB*_Vp_* as a counterselectable marker, we successfully established a novel two-step markerless gene deletion method for *S. suis*. Finally, using the Δ*perR* mutant generated by this method, we demonstrated that *pmtA*, a gene encoding a ferrous iron and cobalt efflux pump in *S. suis* [32] was negatively regulated by the PerR regulator.

## 2. Materials and Methods

### 2.1. Bacterial Strains, Plasmids, Primers, and Culture Conditions

Bacterial strains and plasmids used in this study are listed in Table 1. All primers are listed in Appendix A. Unless otherwise specified, *S. suis* strains were cultured at 37 °C in Tryptic Soy Broth (TSB) or on Tryptic Soy Agar (TSA; Becton, Dickinson and Company, Suzhou, China) supplemented with 5% (vol/vol) newborn bovine serum (Sijiqing, Hangzhou, China). *E. coli* strains were grown in Luria–Bertani (LB) broth or on LB agar. When required, spectinomycin was added to the medium at 50 μg/mL for *E. coli* and 100 μg/mL for *S. suis*.

### 2.2. Preparationof Synthetic Peptide and Natural Transformation Experiment

The peptide (GNWGTWVEE) was synthesized by Sangon Biotech (Shanghai, China) at 90–95% purity. It was dissolved in deionized water at a final concentration of 5 mM, divided into aliquots of 50 μL, and stored at −80 °C.

A natural transformation experiment was performed as previously described [22], with slight modifications. Overnight culture of *S. suis* was diluted 1:100 in fresh medium and grown to an OD_600_ of 0.035–0.05 (about 1–2 h). Next, a 100 μL sample was removed from the culture; 5 μL of the peptide and 1.2 μg of DNA (plasmid or PCR products) were added to the sample. Following 2 h of incubation, the sample was plated on agar plates containing spectinomycin or diluted in fresh media containing 0.5 mM CuSO_4_.

### 2.3. Construction of a S. suis Strain Expressing the Copper-Inducible YoeB_Vp_ Toxin

The promoter P*copA* was amplified from the *S. suis* SC19 genome using primers P*copA*-F/P*copA*-R. The DNA fragment containing *yoeB_Vp_* and its terminator was amplified from *V. parahaemolyticus* RIMD 2210633 genome using primers *yoeB_Vp_*-F/*yoeB_Vp_*-R. The two DNA fragments were fused into a fragment using overlap PCR with primers P*copA*-F/*yoeB_Vp_*-R. Following digestion with the *Bam*H I and *Eco*R I enzymes, the fused DNA fragment was cloned into the pSET2 vector, to generate pSET2-P*copA*-*yoeB_Vp_*. Next, the vector was introduced into the *S. suis* SC19 strain by natural transformation. The resultant strain, SC19/pSET2-P*copA*-*yoeB_Vp_* was confirmed by PCR, DNA sequencing, and reverse transcription quantitative PCR (RT-qPCR).

### 2.4. RNA Extraction

The SC19/pSET2-P*copA*-*yoeB_Vp_* strain was grown to an OD_600_ of 0.6–0.8 and divided into four aliquots of 1 mL, which were supplemented with deionized water or CuSO_4_ at final concentrations of 0.1 mM, 0.2 mM, or 0.5 mM. After 15 min of incubation, bacterial cells were collected and subjected to RNA extraction using an Eastep Super Total RNA Isolation Kit (Promega, Shanghai, China). RNA was evaluated for integrity by gel electrophoresis and determined for concentration using a Nanodrop 200.

In another assay, the WT and Δ*perR* strains were grown to an OD_600_ of 0.6–0.8; each strain was then divided into four aliquots of 1 mL. Three of the aliquots were supplemented with 1 mM FeSO_4_, 1 mM CoSO_4_, and 1 mM NiSO_4_, respectively; the remaining aliquot was supplemented with deionized water. After 15 min of incubation, bacterial cells were collected for RNA extraction. 

### 2.5. RT-qPCR Analysis

cDNA was generated from approximately 0.2 μg of RNA using the NovoScript Plus All-in-one 1st Strand cDNA Synthesis SuperMix (gDNA Purge) (novoprotein, Shanghai, China). Quantitative PCR was performed using NovoStart SYBR qPCR SuperMix Plus (novoprotein, Shanghai, China) and the specific primers listed in Appendix A. The reaction mixture was as follows: 2×NovoStart SYBR qPCR SuperMix Plus 10 μL, each primer 0.5 μM, 10-fold diluted cDNA 1 μL, ROX 0.4 μL, and finally RNase-free water added to 20 μL. Quantitative PCR was conducted on the StepOnePlus Real-Time PCR System (Applied Biosystems). The procedure was 95 °C for 1 min, followed by 40 cycles of 95 °C for 20 s, and 60 °C for 1 min. A melting curve analysis (starting from 60 °C and continuing to 95 °C, with 0.3 °C increments for 5 s each) was performed to verify the specificity of the products. The amplification efficiency of each primer pair was determined using serially diluted genomic DNA as the template. The gene expression level was calculated using the 2^−ΔΔCT^ method [35], with 16S rRNA as the reference gene.

### 2.6. Growth Curves Analyses

Overnight cultures of the SC19/pSET2-P*copA*-*yoeB_Vp_* and SC19/pSET2 strains were diluted in fresh medium and grown to an OD_600_ of approximately 0.3. Next, each culture was divided into five aliquots (1 mL per aliquot), to which CuSO_4_ was added at final concentrations of 0, 0.05, 0.1, 0.2, and 0.5 mM, respectively. Each aliquot was sub-packed in triplicate in 96-well plates (200 μL/well) and cultured at 37 °C for 6 h. The OD_595_ values were measured hourly using the CMax Plus plate reader (Molecular Devices, Shanghai, China).

### 2.7. Spot Dilution Assays

Overnight cultures of the SC19/pSET2-P*copA*-*yoeB_Vp_* and SC19/pSET2 strains were diluted in fresh medium and grown to an OD_600_ of approximately 0.6. Next, each culture was serially diluted 10-fold up to 10^−5^ dilution, and 5 µl of each dilution was spotted onto the plates supplemented with varying concentrations of CuSO_4_ (0, 0.1, 0.2, and 0.5 mM). The plates were photographically documented following 18 h of incubation at 37 °C.

### 2.8. Construction of the SCIY Positive-Negative Selectable Cassette

The spectinomycin-resistance gene was amplified from pSET2 using primers *spc*-F/*spc*-R. The P*copA*-*yoeB_Vp_* construct was amplified from pSET2-P*copA*-*yoeB_Vp_* using primers P*copA*-*yoeB_Vp_*-F/P*copA*-*yoeB_Vp_*-R. The two DNA fragments were fused into a fragment using overlap PCR with primers *spc*-F/P*copA*-*yoeB_Vp_*-R. The fused DNA fragment was confirmed by DNA sequencing, and designated SCIY.

### 2.9. Construction of Markerless Gene Deletion Mutants Using the SCIY Cassette

The Δ*pmtA* mutant was constructed using the SCIY cassette via a two-step procedure. For the first step, the left and right arms of *pmtA* were amplified from *S. suis* SC19 genome using primer pairs *pmtA*-LA-F/*pmtA*-Fir-LA-R and *pmtA*-Fir-RA-F/*pmtA*-RA-R, respectively. The SCIY cassette was amplified using primers *pmtA*-SCIY-F/*pmtA*-SCIY-R. The three DNA fragments were fused into a fragment using overlap PCR with primers *pmtA*-LA-F/*pmtA*-RA-R. The fused DNA fragment was transformed into *S. suis* SC19 by natural transformation. The spectinomycin-resistant colonies were selected, confirmed by PCR, and designated the intermediate strain. For the second step, the left and right arms of *pmtA* were amplified from the *S. suis* SC19 genome using primer pairs *pmtA*-LA-F/*pmtA*-Sec-LA-R and *pmtA*-Sec-RA-F/*pmtA*-RA-R, respectively. The two DNA fragments were fused into a fragment using overlap PCR with primers *pmtA*-LA-F/*pmtA*-RA-R. The fused DNA fragment was transformed into the intermediate strain by natural transformation. Following 2 h of incubation, the sample was diluted 1:100 in fresh medium containing 0.5 mM CuSO_4_ and cultured at 37 °C for another 12 h. In total, the culture was repeatedly diluted three to five times for enrichment of the mutant. After each incubation, the culture was diluted and plated on agar plates. One hundred colonies were tested for spectinomycin-sensitivity. Spectinomycin-sensitive colonies were selected, and the absence of *pmtA* was confirmed by PCR using primer pairs *pmtA*-in-F/*pmtA*-in-R and *pmtA*-out-F/*pmtA*-out-R. The efficiency of the SCIY cassette for counterselection was evaluated as the proportion of spectinomycin-sensitive colonies. The Δ*perR* and Δ*lysR* mutants were constructed using the same procedure to verify the method.

## 3. Results

### 3.1. Identification of the S. suis Strain Expressing the Copper-Inducible YoeB_Vp_ Toxin

To evaluate the effect of YoeB*_Vp_* induction on *S. suis* growth, we constructed a *S. suis* strain expressing the copper-inducible YoeB*_Vp_* toxin using the P*copA* promoter and pSET2 vector [34,36]. The strain, termed SC19/pSET2-P*copA*-*yoeB_Vp_*, was identified by PCR (Figure 1A) and DNA sequencing (data not shown). RT-qPCR analysis was also performed to detect whether copper can induce *yoeB_Vp_* expression. As shown in Figure 1B, the expression of *yoeB_Vp_* was significantly induced by copper, and the inductive effects increased with increasing copper concentrations.

### 3.2. YoeBVp Expression Results in Growth Defect in S. suis

The SC19/pSET2-P*copA*-*yoeB_Vp_* and SC19/pSET2 strains were grown in fresh media containing various concentrations of CuSO_4_, and their growth curves were measured. As shown in Figure 2A, the two strains exhibited similar growth in the absence of CuSO_4_. However, when supplemented with CuSO_4_, the SC19/pSET2-P*copA*-*yoeB_Vp_* strain displayed a remarkable growth defect compared with the SC19/pSET2 strain (Figure 2B–D).

The effect of YoeB*_Vp_* expression on *S. suis* growth was also detected on agar plates. In the absence of CuSO_4_, the two strains formed colonies of equal sizes (Figure 3). However, in the presence of CuSO_4_, the SC19/pSET2-P*copA*-*yoeB_Vp_* strain formed colonies of smaller sizes than the SC19/pSET2 strain (Figure 3).

Taken together, YoeB*_Vp_* expression in *S. suis* led to growth inhibition both in liquid media and on agar plates. Thus, YoeB_Vp_ has the potential to be a counterselectable marker for *S. suis*.

### 3.3. Establishment of the Cloning-Independent and Counterselectable Markerless Gene Deletion System in S. suis

The spectinomycin-resistance gene and P*copA*-*yoeB_Vp_* construct were combined to generate the SCIY cassette, which was further used for markerless gene deletion in *S. suis*. The strategy for markerless gene deletion in *S. suis* using the SCIY cassette is shown in Figure 4. In the first step, an intermediate strain was generated, in which the SCIY cassette replaced the target gene. As the SCIY cassette contains the spectinomycin-resistance gene, the intermediate strain could be selected with spectinomycin. In the second step, the markerless gene deletion mutant was generated. The intermediate strain contains the P*copA*-*yoeB_Vp_* construct; thus, its growth was inhibited in the presence of copper. However, the mutant could grow well in the presence of copper. After three to five dilutions in media supplemented with copper, the mutant was enriched to be easily isolated.

### 3.4. Markerless Deletion of the pmtA, perR, and lysR Genes in S. suis

To assess whether the strategy is effective, we firstly constructed a markerless deletion mutant of the *pmtA* gene. As seen in Figure 5A, PCR amplification of the Δ*pmtA* mutant using primers *pmtA*-in-F/*pmtA*-in-R generated no products, whereas amplification of the WT strain generated products with expected sizes (755 bp). Furthermore, PCR amplification of Δ*pmtA* and the WT strain using primers *pmtA*-out-F/*pmtA*-out-R generated products with expected sizes for Δ*pmtA* (2472 bp) and the WT strain (4199 bp), respectively (Figure 5A). DNA sequencing confirmed that the *pmtA* gene was successfully deleted in the Δ*pmtA* mutant. To further verify the strategy, markerless deletion mutants of the *perR* (Figure 5B) and *lysR* genes (Figure 5C) were also constructed. Overall, the two-step strategy applying the SCIY cassette is effective in markerless gene deletion in *S. suis*.

### 3.5. The SCIY Cassette Is Highly Efficient for Counterselection in S. suis

The proportion of the mutant after each subculture was evaluated to determine SCIY counterselection efficiency in *S. suis*. As shown in Table 2, approximately 95% of the colonies were the Δ*pmtA* mutant after subculturing three times. For the *perR* and *lysR* genes, approximately half or greater than 80% of the colonies were the mutant strain after five times of subculture (Table 2). The results indicate that the SCIY cassette is highly efficient as a counterselectable marker for *S. suis*.

### 3.6. PerR Is a Transcriptional Repressor of the Ferrous Iron and Cobalt Efflux Pump in S. suis

In a previous study, we demonstrated that the *pmtA* gene encodes a ferrous iron and cobalt efflux pump in *S. suis*; its expression was significantly induced by ferrous iron, cobalt, and nickel [32]. Upstream of the *pmtA* gene is a gene encoding the PerR regulator. RT-qPCR analysis was performed to determine whether the *pmtA* gene is under the control of PerR. As shown in Figure 6, *pmtA* expression in the WT strain was upregulated following treatment with ferrous iron, cobalt, and nickel. However, *pmtA* expression in the Δ*perR* mutant was upregulated without metal supplementation (Figure 6). The results reveal that deletion of *perR* led to derepression of the *pmtA* gene; thus, *pmtA* expression in Δ*perR* was upregulated without treatment with ferrous iron, cobalt, or nickel. 

## 4. Discussion

*S. suis* is an important zoonotic pathogen that causes severe infections in swine and humans. Research on the physiology and pathogenesis of *S. suis* usually relies on gene deletion mutants. In the present study, we describe a novel two-step method for markerless gene deletion in *S. suis*. This method is established based on natural transformation in *S. suis* [22] and the utilization of *V. parahaemolyticus* YoeB toxin as a counterselectable marker.

TA systems are widely prevalent in bacteria and archaea [23]. Some toxin genes have been developed as counterselectable markers for genetic manipulation based on toxins’ antibacterial activity [26,27,28,29,30]. In a previous study, induction of *V. parahaemolyticus* YoeB toxin in *E. coli* was found to cause cell death [31]. This finding led to the speculation that YoeB*_Vp_* could be an ideal counterselectable marker. YoeB*_Vp_* expression should be precisely controlled to be an available counterselectable marker. In *S. suis*, the *copA* gene, which encodes a copper efflux system, could be specifically induced by copper [36]. The promoter P*copA* might be reliable to control YoeB*_Vp_* expression in *S. suis*. Herein, a *S. suis* strain expressing the copper-inducible YoeB*_Vp_* toxin was constructed to test the counterselection potential of YoeB*_Vp_*. As expected, the addition of copper to the culture induced *yoeB_Vp_* expression and inhibited *S. suis* growth. It should be noted that a homologous TA system of YefM-YoeB is present in *S. suis* [37]. We also evaluated the counterselection potential of YoeB*_Ss_*. Induction of YoeB*_Ss_* resulted in drastic growth inhibition in *E. coli* [37], whereas no growth defect was observed when YoeB*_Ss_* was induced in *S. suis* (Appendix A). We speculate that the endogenous YefM*_Ss_* antitoxin counteracted the toxicity of YoeB*_Ss_*. While YoeB*_Vp_* shares 63% identity with YoeB*_Ss_* at the amino acid level, YefM*_Vp_* shares only 29% identity with YefM*_Ss_*. Therefore, it is not surprising that the toxicity of YoeB*_Vp_* was not counteracted by YefM*_Ss_*.

A SCIY cassette composed of the spectinomycin-resistance gene and P*copA*-*yoeB_Vp_* construct was generated to explore its application for markerless gene deletion in *S. suis*. The first step, by which the SCIY cassette replaced the target gene, was adopted from a previously described method [22]. The intermediate strain was easily selected from plates containing spectinomycin. Since YoeB*_Vp_* toxin exerts a bacteriostatic effect rather than a bactericidal effect on *S. suis*, the mutant generated from the second step should not be selected directly from plates containing copper. Instead, several dilutions in media containing copper were performed for the enrichment of the mutant. Our results showed that after subculturing three to five times, the mutant was easy to isolate. However, the mutant’s proportion after each subculture should be correlated with the efficiency of natural transformation and homologous recombination.

In a previous study, a cassette containing a kanamycin resistance gene and a gene encoding the ParE toxin has been used to introduce a single mutation in *Salmonella* Typhimurium [30]. Similarly, the SCIY cassette could be applied in site-directed mutagenesis or deletion of a few bases in the genome of *S. suis*, which is an outstanding advantage of the two-step method. The conventional method using pSET4s generates the mutant and WT genotype simultaneously, which were preliminarily identified by PCR. It would be difficult to distinguish the mutant and WT strains by PCR when only a few bases were deleted. If using the two-step method, the SCIY cassette in the intermediate strain could be easily replaced by the target gene with desired site-directed mutagenesis or a few bases deletion in the second step. The two-step method would facilitate research of the role of a single amino acid or protein domain in *S. suis*.

Although the two-step method is highly efficient in markerless gene deletion in *S. suis*, it does not mean that it could not be further improved. Next, the effect of other toxins on *S. suis* growth will be evaluated. If a toxin is found to exert bactericidal activity against *S. suis*, the *yoeB_Vp_* gene in the SCIY cassette will be replaced by this gene. Then, the intermediate strain is expected to be killed in the presence of copper, so that the mutant can be easily isolated in the second step without enrichment. In addition, some undesired mutations might be introduced into the genome during construction of the mutant. Therefore, it would be better to generate a complementation strain for the mutant when performing a functional study of a gene.

BlastP analysis also revealed that YefM*_Vp_* and YoeB*_Vp_* share 30% and 63% amino acid sequence identity with the homologous antitoxin and toxin from *Streptococcus pneumoniae*, respectively [38]. It is likely that the YoeB*_Vp_* toxin could exert a toxic effect against S. *pneumoniae*, which might not be counteracted by the endogenous YefM*_Sp_* antitoxin. Therefore, further studies could be performed to detect the counterselection potential of the YoeB*_Vp_* toxin in other species such as *S. pneumoniae*. Yet, a suitable promoter should be selected to control *yoeB_Vp_* expression in the corresponding species.

In a previous study, we demonstrated that the *pmtA* gene is involved in ferrous iron and cobalt efflux in *S. suis* [32]. One of the remaining questions is which regulator modulates *pmtA* expression. In *Streptococcus pyogenes*, the PmtA homolog is regulated by PerR [39,40]. In *S. suis*, the *perR* gene is located upstream of the *pmtA* gene. Using the Δ*perR* mutant generated by the novel two-step method, we demonstrated that in the absence of metal supplementation, *pmtA* expression in the Δ*perR* mutant was significantly upregulated compared to that in the WT strain. This result is consistent with the observations in *S. pyogenes* [39,40]. Thus, PerR is a transcriptional repressor of *pmtA* in *S. suis*.

In conclusion, a novel two-step markerless gene deletion method was established for *S. suis*. This method is cloning-independent and can also be used for site-directed mutagenesis or deletion of a few bases in the genome of *S. suis*. Moreover, we demonstrate that PerR is a transcriptional repressor of ferrous iron and cobalt efflux pump (PmtA) in *S. suis*.

## Figures and Tables

**Figure 1 microorganisms-09-01095-f001:**
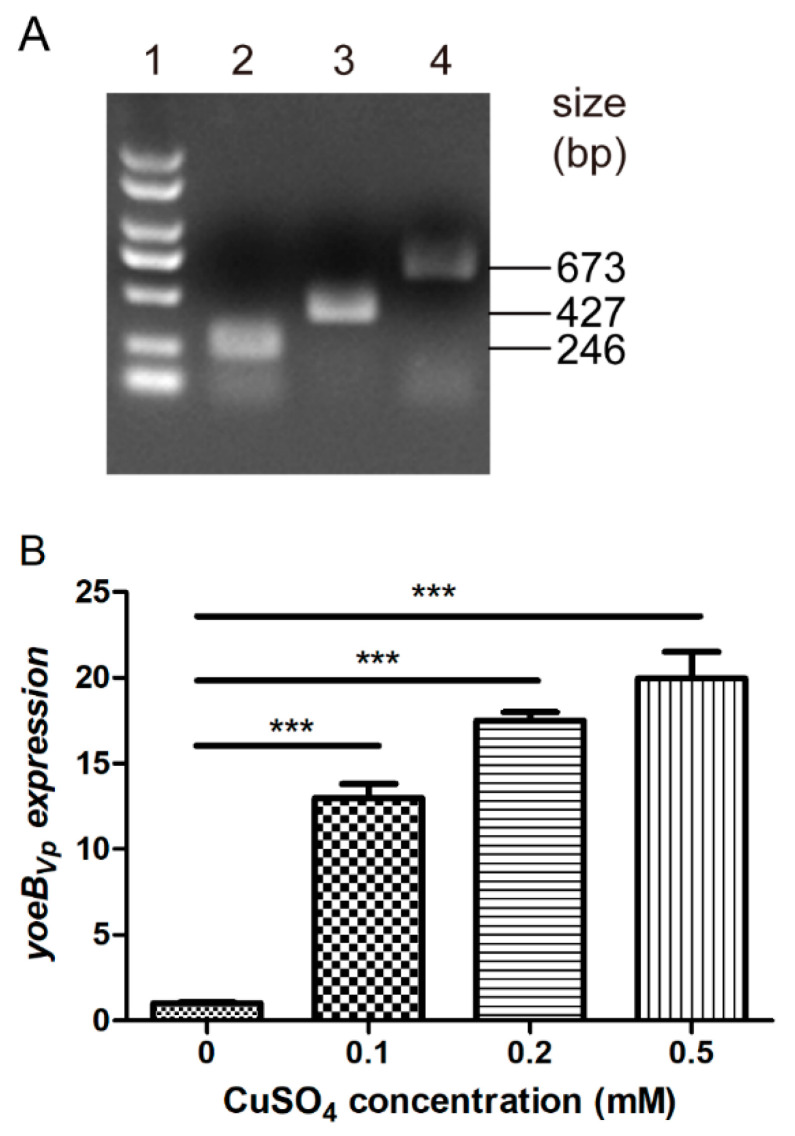
Identification of the *S. suis* strain expressing the copper-inducible YoeB*_Vp_* toxin. (**A**) PCR identification of the SC19/pSET2-P*copA*-*yoeB_Vp_* strain. Lane 1 indicates the DL2000 DNA marker. Lanes 2–4 indicate PCR amplification of the P*copA* promoter, the *yoeB_Vp_* gene, and the P*copA*-*yoeB_Vp_* construct, respectively. (**B**) RT-qPCR identification of the SC19/pSET2-P*copA*-*yoeB_Vp_* strain. The data shown are the means and standard deviations (SD) from three independent experiments. One-way analysis of variance with Bonferroni’s post-test was used for statistical analyses. *** indicates *p* < 0.001.

**Figure 2 microorganisms-09-01095-f002:**
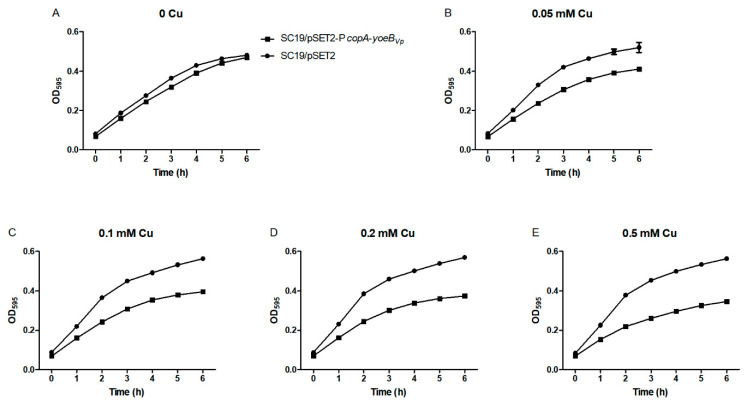
YoeB*_Vp_* expression resulted in a growth defect in *S. suis* in liquid media. The SC19/pSET2-P*copA*-*yoeB_Vp_* and SC19/pSET2 strains were grown in the absence (**A**) and presence of various concentrations of CuSO_4_ (**B**–**E**); (**B**) 0.05 mM CuSO_4_; (**C**) 0.1 mM CuSO_4_; (**D**) 0.2 mM CuSO_4_; (**E**) 0.5 mM CuSO_4_. At least three independent experiments were performed; the data shown are the means ± SDs from three wells in a representative experiment.

**Figure 3 microorganisms-09-01095-f003:**
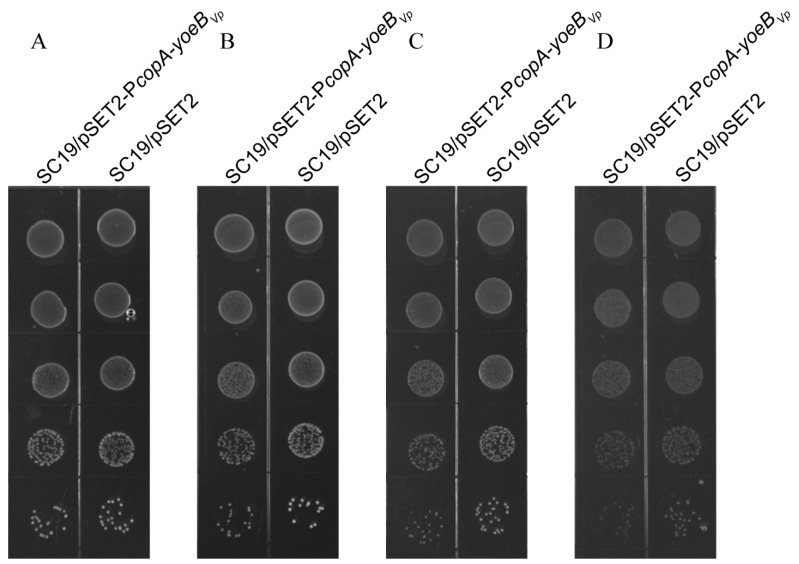
YoeB*_Vp_* expression resulted in growth defect in *S. suis* on agar plates. The SC19/pSET2-P*copA*-*yoeB_Vp_* and SC19/pSET2 strains were grown in the absence (**A**) and presence of various concentrations of CuSO_4_ (**B**–**D**); (**B**) 0.1 mM CuSO_4_; (**C**) 0.2 mM CuSO_4_; (**D**) 0.5 mM CuSO_4_. The images are representative of at least three independent experiments.

**Figure 4 microorganisms-09-01095-f004:**
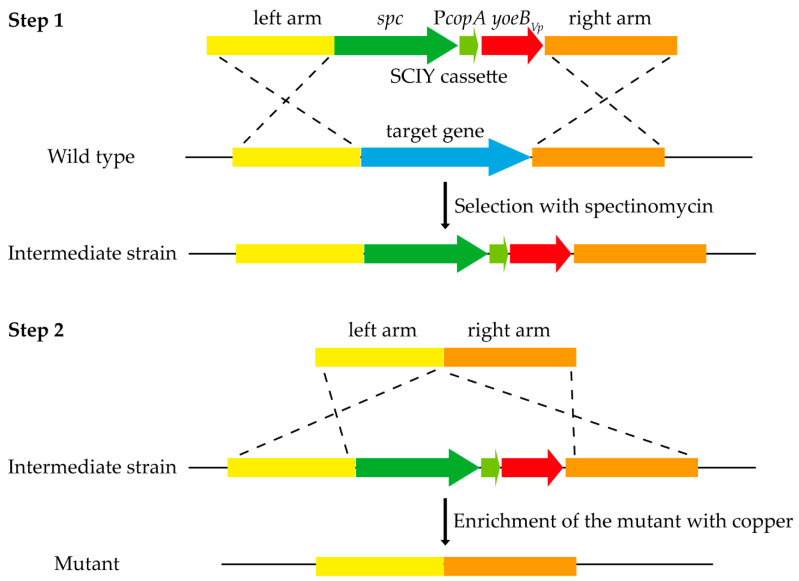
Schematic representation of the two-step markerless gene deletion method in *S. suis*. In the first step, the target gene was replaced by the SCIY cassette. The intermediate strain was selected with spectinomycin. In the second step, the markerless gene deletion mutant was generated. Growth of the intermediate strain was inhibited by copper, whereas the mutant was tolerant to copper. The medium was supplemented with copper for the enrichment of the mutant.

**Figure 5 microorganisms-09-01095-f005:**
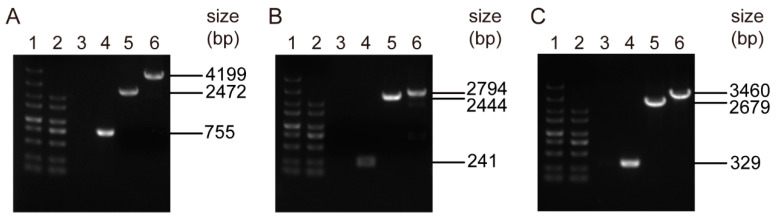
PCR identification of the mutants. (**A**) Δ*pmtA*; (**B**) Δ*perR*; (**C**) Δ*lysR*. Lanes 1 and 2 indicate the DL5000 and DL2000 DNA markers, respectively. Lanes 3 and 5 indicate PCR amplification of the mutants using the primer pairs in-F/in-R and out-F/out-R, respectively. Lanes 4 and 6 indicate PCR amplification of the WT strain using the primer pairs in-F/in-R and out-F/out-R, respectively.

**Figure 6 microorganisms-09-01095-f006:**
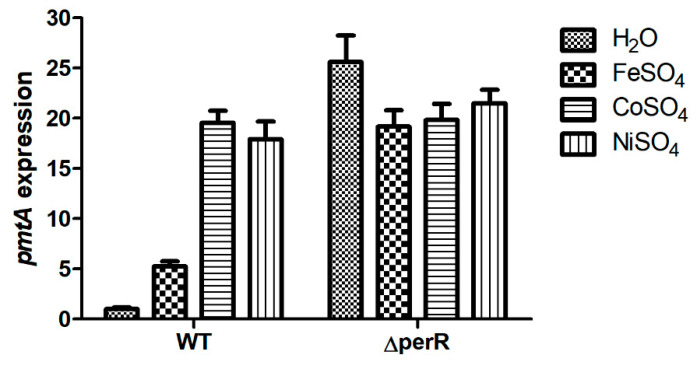
*pmtA* expression in the WT and Δ*perR* strains in the absence and presence of ferrous iron, cobalt, and nickel. The data shown are the means and standard deviations (SD) from three independent experiments.

**Table 1 microorganisms-09-01095-t001:** Bacterial strains and plasmids used in this study.

Strain or Plasmid	Relevant Characteristics ^1^	Source or Reference
Strains		
SC19	Virulent *S. suis* strain isolated from the brain of a dead pig	[33]
SC19/pSET2-P*copA*-*yoeB_Vp_*	Strain SC19 expressing the copper-inducible YoeB*_Vp_* toxin	This study
SC19/pSET2-P*copA*-*yoeB_Ss_*	Strain SC19 expressing the copper-inducible YoeB*_Ss_* toxin	This study
SC19/pSET2	Strain SC19 carrying the pSET2 vector	This study
Δ*pmtA*	*pmtA* deletion mutant of strain SC19	This study
Δ*perR*	*perR* deletion mutant of strain SC19	This study
Δ*lysR*	*lysR* deletion mutant of strain SC19	This study
MC1061	Cloning host for recombinant vector	AngYuBio, Shanghai, China
Plasmids		
pSET2	*E. coli-S. suis* shuttle vector; Spc^R^	[34]
pSET2-P*copA*-*yoeB_Vp_*	pSET2 containing the *yoeB_Vp_* gene and P*copA* promoter	This study
pSET2-P*copA*-*yoeB_Ss_*	pSET2 containing the *yoeB_Ss_* gene and P*copA* promoter	This study

^1^ Spc^R^, spectinomycin resistant.

**Table 2 microorganisms-09-01095-t002:** The proportion of spectinomycin-sensitive colonies (mutants).

Gene	Repetition	Spectinomycin-Sensitive Colonies (Mutants) (%) ^1^				
		First Dilution	Second Dilution	Third Dilution	Fourth Dilution	Fifth Dilution
*pmtA*	Rep_1	5	75	98		
	Rep_2	3	91	98		
	Rep_3	1	65	95		
*perR*	Rep_1	0	0	63	57	49
	Rep_2	0	32	98	100	98
	Rep_3	0	5	83	93	97
*lysR*	Rep_1	2	1	27	24	86
	Rep_2	0	1	0	15	52
	Rep_3	2	0	1	32	87

^1^ The percentage of spectinomycin-sensitive colonies (mutants) was determined by analysis of 100 colonies. The experiments were performed three times for deletion of each gene, and the results for each repetition are shown.

## Data Availability

Not applicable.

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
