# Peer review of "A Markerless Gene Deletion System in *Streptococcus suis* by Using the Copper-Inducible *Vibrio parahaemolyticus* YoeB Toxin as a Counterselectable Marker"

_microorganisms, 2021, doi:10.3390/microorganisms9051095_

Round 1
Reviewer 1 Report
A nice and well documented methods manuscript.
Minor comments:
In the Summary and Introduction, the sentence " we demonstrated that the pmtA gene, which encodes a ferrous iron and cobalt efflux pump in S. suis, was negatively regulated by the PerR regulator", should include that the described markerless deletion method is used.
Table 2: "primers used in this study" can be moved to Supplementary Material.
Author Response
Thank you very much for your appreciation of our manuscript. The manuscript has been revised according to your suggestions. The following response is point-by-point towards the comments.
In the Summary and Introduction, the sentence " we demonstrated that the pmtA gene, which encodes a ferrous iron and cobalt efflux pump in S. suis, was negatively regulated by the PerR regulator", should include that the described markerless deletion method is used.
A: Thank you very much for your suggestion. The sentence has been rephrased in the revised manuscript. Please refer to lines 26, 80-81, 369-370.
Table 2: "primers used in this study" can be moved to Supplementary Material.
A: Table 2 has been moved to Supplementary Material as suggested.
Reviewer 2 Report
The manuscript by Zheng et al presents a novel method of gene deletion in Streptococcus suis using the toxin gene from Vibrio parahaemolyticus YoeB to select for the marker-less clones after the recombination. The proposed system is based on the copper-induced promoter controlling toxin expression. In general, this two-steps system should improve cloning and constructing deletion mutants in S. suis, the economically important swine pathogen, thus, the subject is worth pursuing and the proposed method attractive. However, there are several issues that require clarification or additional explanation and modifications.
Specific issues are as follows:
- the most important question is whether the proposed cloning system is – in addition to be marker-less – also seamless. What are the results of the sequencing of the mutants?
- more detailed information on the RT-qPCR analysis is needed such as validation of reference genes, normalization of Ct, reverse transcription conditions and repetitions
- the last paragraph of the introduction (ln 68-78) is a extensive summary of the paper’s findings, and should be shortened to 1-2 sentences to avoid repetitions
- table 2 could be transferred to the supplementary data
- the authors claimed that 0.5 mM CuSO4 has a most prominent effect in expressing yoeB gene (Fig 1 and ln 180), however perhaps higher concentrations of copper would give better effect?
- Figure 2 – the layout of the figure prevents from the direct comparison of the effects of increasing copper concentration
- the growth inhibition showed on the solid medium does not look in concert with the experiments performed in the liquid medium (Fig 2 and 3) – is the only effect of toxin overexpression showed in the colony size, not CFU?
- while referring to the cloning steps, the strain with the spect. cassette should be called “intermediate” rather than “middle”
- the entire paragraph on PerR (3.6) looks somehow out of the line of the paper. While choosing the specific genes as examples of the cloning is justified, the extended analysis of the lack of one of the regulators should be presented in the separate story
- what was a rationale behind choosing the V. parahemolyticus toxin instead of S. suis own toxins?
- I would not agree that this system is cloning-independent (ln 335) as certain cloning steps are required
- The language requires correction in both grammar and proper using of the certain terms
Author Response
Thank you very much for your positive comments and valuable suggestions on our manuscript, which help us in depth to improve the quality of our manuscript. We have tried our best to revise the manuscript according to the suggestions. The following response is point-by-point towards the comments.
- the most important question is whether the proposed cloning system is – in addition to be marker-less – also seamless. What are the results of the sequencing of the mutants?
A: Yes. The novel gene deletion system is seamless, which has been demonstrated by DNA sequencing of the mutants (lines 266). Actually, this system is still based on homologous recombination. Gene deletion methods based on homologous recombination have been described in S. suis and other bacterial species. Please refer to references [1-5].
- more detailed information on the RT-qPCR analysis is needed such as validation of reference genes, normalization of Ct, reverse transcription conditions and repetitions
A: Thank you very much for your suggestion. More detailed information on the RT-qPCR analysis has been added. Please refer to lines 132-140 in the revised manuscript.
- the last paragraph of the introduction (ln 68-78) is an extensive summary of the paper’s findings, and should be shortened to 1-2 sentences to avoid repetitions
A: This paragraph has been shortened; please refer to lines 71-82 in the revised manuscript.
- table 2 could be transferred to the supplementary data
A: Table 2 has been transferred to the supplementary data.
- the authors claimed that 0.5 mM CuSO4 has a most prominent effect in expressing yoeB gene (Fig 1 and ln 180), however perhaps higher concentrations of copper would give better effect?
A: Thank you for pointing out our negligence. In a previous study, we found that 1 mM CuSO4 could inhibit S. suis growth [6]. Therefore, we selected copper concentrations lower than 1 mM for gene expression and growth curve analyses. The sentence has been rephrased in the revised manuscript. Please refer to lines 191-193.
- Figure 2 – the layout of the figure prevents from the direct comparison of the effects of increasing copper concentration
A: Thank you for pointing out our negligence. We have checked the raw data again, and found there is only a small dose-dependent effect of copper concentrations. The sentence has been rephrased in the revised manuscript. Please refer to lines 219-220.
- the growth inhibition showed on the solid medium does not look in concert with the experiments performed in the liquid medium (Fig 2 and 3) – is the only effect of toxin overexpression showed in the colony size, not CFU?
A: Actually, the growth inhibition in liquid media is consistent with that on agar plates. Growth inhibition in liquid media might be due to either bactericidal effect (resulted in less CFU and smaller colony size) or bacteriostatic effect (resulted in smaller colony size). Our results revealed that the YoeBVp toxin exerts a bacteriostatic effect towards S. suis.
- while referring to the cloning steps, the strain with the spect. cassette should be called “intermediate” rather than “middle”
A: Thank you very much for your suggestion. Correction has been made as suggested. Please refer to lines 170, 174, 243, 245, 246, 255, 256, 331, 346, 353 and Figure 4.
- the entire paragraph on PerR (3.6) looks somehow out of the line of the paper. While choosing the specific genes as examples of the cloning is justified, the extended analysis of the lack of one of the regulators should be presented in the separate story
A: It is true that the regulation of PmtA by PerR looks out of the line of the paper. Actually, we have characterized the PmtA system in a previous study [7], and the PerR regulator has been studied by another group [8]. The regulation of PmtA is a remaining question. In this study, we just constructed the pmtA and perR mutants to verify the novel gene deletion method; we also hope to give an answer to the remaing question. As suggested by another reviewer, several sentences have been rephrased to make this result more relevant. Please refer to lines 26, 80, and 369-370.
- what was a rationale behind choosing the V. parahemolyticus toxin instead of S. suis own toxins?
A: Actually, we also assessed the S. suis YoeB toxin. While induction of YoeBSs resulted in remarkable growth inhibition in E. coli, no growth inhibition was observed in S. suis. We speculate that YoeBSs was counteracted by the S. suis YefM antitoxin. Discussion about it has been added in the revised manuscript; please refer to lines 331-324.
- I would not agree that this system is cloning-independent (ln 335) as certain cloning steps are required
A: Usually, gene cloning means that a DNA fragment ligates into a vector. This system works without a vector. The DNA fragment was generated by PCR amplification. Thus, it is cloning-independent. Please refer to reference [4] for the same description.
- The language requires correction in both grammar and proper using of the certain terms
A: Thank you very much for your suggestion. The manuscript has been edited by Wordvice language editing service. In addition, the manuscript will be checked for grammar and terms by the Editorial Office if it is accepted.
References:
[1] Takamatsu D, et al. Thermosensitive suicide vectors for gene replacement in Streptococcus suis. Plasmid 2001, 46, 140-148.
[2] Zaccaria E, et al. Control of competence for DNA transformation in Streptococcus suis by genetically transferable pherotypes. PLoS One 2014, 9(6):e99394.
[3] Zhu Y, et al. Utilization of the ComRS system for the rapid markerless deletion of chromosomal genes in Streptococcus suis. Future Microbiol 2019, 14:207-222.
[4] Xie Z, et al. Cloning-independent and counterselectable markerless mutagenesis system in Streptococcus mutans. Appl Environ Microbiol 2011, 77(22):8025-8033.
[5] Song P, et al. Scarless gene deletion in methylotrophic Hansenula polymorpha by using mazF as counter-selectable marker. Anal Biochem 2015, 468:66-74.
[6] Zheng C, et al. CopA protects Streptococcus suis against copper toxicity. Int J Mol Sci 2019, 20(12):2969.
[7] Zheng C, et al. PmtA functions as a ferrous iron and cobalt efflux pump in Streptococcus suis. Emerg Microbes Infect 2019, 8, 1254-1264.
[8] Zhang T, et al. A Fur-like protein PerR regulates two oxidative stress response related operons dpr and metQIN in Streptococcus suis. BMC Microbiol 2012, 12:85.
Reviewer 3 Report
The work by Zheng et al., describes a new method to make scarless deletions in Streptococcus suis, an important zoonotic pathogen. To this end, authors combined natural transformation, the toxicity of YoeB from Vibrio parahaemolyticus and a tight yoeB gene expression regulation by the copper-inducible promoter (PcopA).
I believe this article can be a valuable tool for researchers in the field. The development of new technology to engineer bacterial species can greatly advance the understanding of S. suis and perhaps related species.
I have a few comments on the text, figures and data that I have included in the attached PDF file.
In addition I have some questions / suggestions for the Discussion:
- Would the usage of the endogenous YoeB from S. suis provide a higher counter selection?.
- Could the authors comment on the possibility of adapting the SCIY cassette to other Streptococcus species?
- Lines 313-320. Authors could comment on a previous 2 marker-selection methods used to introduced single mutations in the genome (Lobato-Marquez et al., Fron Mol Biosci 2016).
- The usage of counter selection markers may have associated undesired compensatory mutations in the genomes. Could the authors comment on this?

Author Response
Thank you very much for your positive comments and valuable suggestions on our manuscript, which help us in depth to improve the quality of our manuscript. We have tried our best to revise the manuscript according to the suggestions. The following response is point-by-point towards the comments.
Line 62. There are more recent TA reviews: e.g. Diaz-Orejas et al., Front Microbiol 2017; Harms et al., Mol Cell 2018
A: The references has been replaced by the recent TA reviews. Please refer to lines 65 and references 24-25 in the revised manuscript.
Line 64. Authors are missing examples where TA systems have been effectively used as counterselectable markers; e.g. ParE in Lobato-marquez et al., front mol biosci 2016. Actually in this study ParE was used not only as a counterselection marker to design bacterial mutants, but also to select for plasmid-less bacteria in an efficient way.
A: Yes. ParE is an important example of counterselection marker. It has been added in the revised manuscript. Please refer to lines 67 and reference 30.
Line 88 (Table 1). This (S. suis) should be in italics; “YoeB” Authors are referring to a gene so this should be italics
A: Thank you for pointing out our negligence. Correction has been made for S. suis. YoeB are referring to the YoeB toxin; it is a protein.
Line 107. What enzymes? Please specify
A: The enzymes have been specified. Please refer to line 111.
Line 117. “The Nanodrop 200 instrument” to “a Nanodrop 200”
A: Thank you for pointing out our negligence. Correction has been made as suggested. Please refer to line 122.
Line 197. Authors should specify:
1- What is represented; e.g. mean +- SEM
2- the statistical analysis used
3- how many times this experiment was performed.
A: Thank you very much for your suggestions. The data shown are the means and standard deviations (SD) from three independent experiments. One-way analysis of variance with Bonferroni’s post-test was used for statistical analyses. The information has been added. Please refer to lines 211-213.
Lines 207-208. Even though from the images it really looks like there is a growth difference between pSET2 and pSET2-PcopA-yoeB carrying strains, the reader would benefit from a cropped imaged where the colonies are magnified so the difference is size become more obvious.
A: Thank you for your suggestion. We have tried to provide magnified images, but found that they appeared a bit pixelated. The current images clearly showed that the strain carrying pSET2-PcopA-yoeB had a growth defect compared with the strain carrying pSET2.
Line 213 and Figure 2. Would it be worthy to show all growth curves in a single graph? That way it would be easier to compare the dose-dependent effect of copper and subsequent YoeB toxicity.
A: Thank you for pointing out our negligence. We have checked the raw data again, and found there is only a small dose-dependent effect of copper concentrations. The sentence has been rephrased in the revised manuscript. Please refer to lines 219-220.
Line 218. I have a question regarding how the spot assay was performed. Normally for these assays all serially diluted aliquots are plated in the same plate with a defined concentration of the inducer and all dilutions imaged together. Was this the way the authors imaged the samples? I ask because it looks like not all spots come from the same plate? Did the authors use petri dishes or multi-well plates.
A: We used petri dishes in this assay. The agar plates were prepared with various concentrations of Cu. For each independent experiment, all serially diluted aliquots are plated in four plates, which were supplemented with varying concentrations of CuSO4 (0, 0.1, 0.2, and 0.5 mM). Please refer to lines 151-155.
Line 220. I would call this images and not graphs
A: Thank you for pointing out our negligence. Correction has been made as suggested. Please refer to line 236.
Line 265. “were” to “are”
A: Thank you for pointing out our negligence. Correction has been made as suggested. Please refer to line 285.
Lines 277-280. This Figure legend seems to be wrong (it is the figure legend from figure 4).
A: Thank you for pointing out our negligence. The figure legend has been replaced in the revised manuscript. Please refer to lines 300-302.
Would the usage of the endogenous YoeB from S. suis provide a higher counter selection?
A: Actually, we also assessed the usage of the endogenous S. suis YoeB. While induction of YoeBSs resulted in remarkable growth inhibition in E. coli, no growth inhibition was observed in S. suis. We speculate that the toxicity of YoeBSs was counteracted by the endogenous S. suis YefM antitoxin. Discussion about it has been added in the revised manuscript; please refer to lines 321-324.
Could the authors comment on the possibility of adapting the SCIY cassette to other Streptococcus species?
A: Thank you very much for your suggestion. The possibility of adapting the YoeBVp toxin in other Streptococcus species has been discussed in the revised manuscript. The promoter might need to be replaced in other species. Please refer to lines 359-365.
Lines 313-320. Authors could comment on a previous 2 marker-selection methods used to introduced single mutations in the genome (Lobato-Marquez et al., Fron Mol Biosci 2016).
A: Thank you very much for your suggestion. Comments on the method has been added in the revised manuscript. Please refer to lines 339-341.
The usage of counter selection markers may have associated undesired compensatory mutations in the genomes. Could the authors comment on this?
A: Yes. Any gene deletion methods have this problem. Therefore, the complementation strain should be constructed and included in subsequent experiments. Comment on this has been added in the revised manuscript. Please refer to lines 355-358.
Round 2
Reviewer 2 Report
The authors successfully answered all questions and issues so in my opinion the manuscript is ready to be published